# Fixed-Distance Hamiltonian Monte Carlo

**Hadi Mohasel Afshar**
CSIRO's Data61
Eveleigh, NSW 2015, Australia
Hadi.Afshar@data61.csiro.au

**Sally Cripps**
CSIRO's Data61 & The University of Sydney
Eveleigh, NSW 2015, Australia
Sally.Cripps@data61.csiro.au

## Abstract

We propose a variation of the Hamiltonian Monte Carlo sampling (HMC) where the equations of motion are simulated for a fixed traversed distance rather than the conventional fixed simulation time. This new mechanism tends to generate proposals that have higher target probability values. The momentum distribution that is naturally joint with our Fixed-Distance HMC (FDHMC), and keeps the proposal acceptance probability close to 1, is not Gaussian and generates momentums that have a higher expected magnitude. This translates into a reduced correlation between the successive MCMC states and according to our experimental results, leads to an improvement in terms of the effective sample size per gradient when compared to the baseline HMC and No-U-Turn (NUTS) samplers.

## 1 Introduction

Markov chain Monte Carlo (MCMC) is an inference mechanism that approximates a target probability distribution by a sequence of *states* (a.k.a. *samples* or *draws*) (Chib and Greenberg, 1995). Hamiltonian Monte Carlo (HMC) (Duane et al., 1987; Neal, 2011) is an MCMC algorithm where the current state is considered as a particle (usually a unit mass) to which a randomly drawn momentum vector is assigned. The state is also associated with a potential energy that is proportional to its negative log target probability density. The next state is proposed via evolving the current state according to a simulation of the equations of motion (using leapfrog integration) for a fixed time duration. If the simulation is accurate, the probability of accepting the proposal as the next state in the Markov chain will be close to one. This property makes HMC a powerful sampling tool where the proposal can be quite distant from the current state while being accepted with a high probability, leading to a comparatively rapid exploration of the target distribution. In this work we further improve on HMC by highlighting two of its limitations and resolving them.

*Limitation 1. Along the trajectory that is traversed during any fixed-time simulation of the equations of motion, it is more likely that a state with low target probability is proposed:* In HMC, the state is guided by the gradient information of the target density to move towards regions of high probability (i.e. low potential engery). However, it moves through these regions with a high velocity and thus oscillates around them, spending more time in the surrounding low-probability regions (i.e. hills on the potential energy) where it moves with low velocity. As such, it is more likely that the state is in a low-probability region when the simulation time is over. This bias is counter intuitive.

What makes the overall sampling process unbiased (in the sense that the MCMC chain converges to the target distribution) relies on the HMC's momentum distribution: Note that a state that has a low-magnitude momentum cannot move to a low-probability region (since at some level, its kinetic energy will be zero, the direction of the momentum vector will be reversed and the state will be pulled back) but a state with low-magnitude momentum can always move from a low-probability region to a high-probability region as its kinetic energy will only be increased. Therefore, if the momentum distribution is concentrated around $\mathbf{0}$, there is a strong bias towards high-probability regions. Conversely, if the momentum distribution favours large magnitudes, the bias is less pronounced. The exact distribution

36th Conference on Neural Information Processing Systems (NeurIPS 2022).

that generates "right amount of low-magnitude momentums" to counterbalance the bias that is induced by Limitation 1, happens to be a **0**-mean Gaussian. However using this distribution to generate low-magnitude momentums causes another limitation for HMC:

*Limitation 2. HMC suffers from correlated samples in regions of high probability:* A state that has a low-magnitude momentum, cannot leave a high-probability region (as it does not have sufficient kinetic energy to lift it out of the potential energy valley). During the simulation time, it only oscillates (or shivers) in its place leading to a proposal that is not distant from the current state. Therefore, despite HMC's nominal high proposal acceptance probability, it can also produce highly correlated close-by samples.

Note that algorithms such as *No-U-turn sampling* (NUTS) (Hoffman and Gelman, 2014) where the number of simulation leapfrog steps, and hence the simulation duration, is dynamically decided, do not resolve *Limitation 1*, because regardless of the number of the leapfrogs, or whether the state has made a U-turn or not, high-probability regions are traversed faster than low probability regions (and less intermediate states are generated from them) and therefore they are under-represented. Regarding *Limitation 2*, a dynamically decided simulation duration, may save some computation by avoiding redundant oscillations for a state that is trapped in an energy valley, but it does not guarantee that the generated proposal is distant from the current state and hence limitation 2 is not addressed either.

The contribution of the present work is to systematically tackle both aforementioned HMC limitations by simulating the equations of motion for a fixed traversed distance rather than a fixed evolution time. An evolving state may spend a longer time in low-probability regions but due to its low momentum/velocity, the traversed distance (in each leapfrog step) is short. Conversely, even though less time may be spent in high-probability regions, due to the higher momentum, the traversed distance (per leapfrog step) is longer. Therefore the fixed distance budget is exhausted more rapidly in high-probability regions and the bias against these regions disappears.

This modification to the HMC algorithm also resolves Limitation 2. Intuitively speaking, no counterbalancing is required because there is no bias towards regions of low probability in our Fixed-Distance HMC (FDHMC). As such, in order to maintain the proposal acceptance probability close to 1, FDHMC's momentum distribution needs to generate values with higher expected magnitude than those generated from a Gaussian centred at **0**. This modification translates into a faster exploration of the probability space and substantially less correlated samples.

To design FDHMC, we rely on the theory of Reversible Jump MCMC (RJMCMC) (Green, 1995). There are recent works that see HMC in the light of RJMCMC (Levy et al., 2018; Afshar et al., 2021) to design non-volume-preserving extensions. But we are unaware of any existing variation of HMC where the traversed distance is fixed, or where the momentum vectors are drawn from a distribution other than normal, or where any solution is proposed for the two aforementioned limitations. After providing a brief background in Section 2 we will design FDHMC in sections 3 and 4. According to our experiments that are provided in Section 5, FDHMC can outperform the baseline HMC and NUTS by a substantial margin, in terms of the effective sample size per gradient.

## 2 Background

Our ultimate task is to approximate a *target density*, $\pi_{\mathbf{Q}}(\mathbf{q})$, defined on $\mathbb{R}^n$, with a set of *states* $\{\mathbf{q}^{(c)}\}_{c=1}^N$ (also known as *samples* or *draws*). In Markov chain Monte Carlo (MCMC) framework, each sample, $\mathbf{q}^{(c)}$, is drawn conditioned on the previous sample, $\mathbf{q}^{(c-1)}$. Nonetheless, the approximation is *unbiased*. That is, if the number of draws $N \rightarrow \infty$, then the number of draws that are within any region, $\mathcal{A} \subset \mathbb{R}^n$, is proportional to the probability mass associated with that region, $\int_{\mathcal{A}} \pi_{\mathbf{Q}}(\mathbf{q})d\mathbf{q}$.

*Reversible Jump MCMC* (Green, 1995) is an MCMC technique where to draw the next state, $\mathbf{q}^{(c+1)}$:
(a) The current state, $\mathbf{q}^{(c)}$, is augmented by an *auxiliary vector*, $\mathbf{r}^{(c)}$, that is drawn from an arbitrary *auxiliary distribution* (that throughout, we assume, is continuous with a density function , $\pi_{\mathbf{R}}(\mathbf{r})$, that is defined on $\mathbb{R}^m$, for some dimensionality $m$).
(b) $(\mathbf{q}^{(*)}, \mathbf{r}^{(*)}) := \mathcal{F}(\mathbf{q}^{(c)}, \mathbf{r}^{(c)})$ is proposed as the next augmented state where $\mathcal{F} : \mathbb{R}^{n+m} \rightarrow \mathbb{R}^{n+m}$ is an arbitrary differentiable mapping that is *reversible*, in the sense that $\mathcal{F}$ is the inverse of itself:

$$\mathcal{F}(\mathbf{q}, \mathbf{r}) = (\mathbf{q}', \mathbf{r}'), \quad \text{if and only if,} \quad \mathcal{F}(\mathbf{q}', \mathbf{r}') = (\mathbf{q}, \mathbf{r}). \qquad \text{(Reversibility condition)} \quad (1)$$

(c) With Metropolis-Hastings-Green (MHG) acceptance probability:

$$\alpha\left((\mathbf{q}^{(c)}, \mathbf{r}^{(c)}) \to (\mathbf{q}^{(*)}, \mathbf{r}^{(*)})\right) := \min\left\{1, \frac{\pi_{\mathbf{Q}}(\mathbf{q}^{(*)})\pi_{\mathbf{R}}(\mathbf{r}^{(*)})}{\pi_{\mathbf{Q}}(\mathbf{q}^{(c)})\pi_{\mathbf{R}}(\mathbf{r}^{(c)})} \cdot \left|\frac{\partial(\mathbf{q}^{(*)}, \mathbf{r}^{(*)})}{\partial(\mathbf{q}^{(c)}, \mathbf{r}^{(c)})}\right|\right\}, \tag{2}$$

the proposal, $(\mathbf{q}^{(*)}, \mathbf{r}^{(*)})$, is accepted as the next augmented MCMC state, $(\mathbf{q}^{(c+1)}, \mathbf{r}^{(c+1)})$, otherwise the current state, $(\mathbf{q}^{(c)}, \mathbf{r}^{(c)})$, is returned.

(d) $\mathbf{q}^{(c+1)}$ is kept (as the next draw from the target density) and the auxiliary $\mathbf{r}^{(c+1)}$, is discarded.

*Hamiltonian Monte Carlo* (HMC) (Duane et al., 1987) is a special case of RJMCMC where (a) the original state, $\mathbf{q}$, (that in this context, is referred to as the *position* vector) and the auxiliary vector (that is referred to as the *momentum* vector and is usually denoted by $\mathbf{p}$) have the same dimensions (i.e. $n = m$). (b) The auxiliary density function, $\pi_{\mathbf{P}}(\mathbf{p})$, is an $n$-variate normal distribution which is usually assumed to be standard, $\mathcal{N}(\mathbf{0}_n, \mathbf{I}_{n \times n})$. (c) The reversible mapping, $\mathcal{F}$, is the simulation of the equations of motion for the total time $\epsilon L$, via *momentum-Verlet* integration with $L$ leapfrog steps, each of duration $\epsilon$, followed by reversing the sign of the momentum vector:

$$\mathcal{F}(\mathbf{q}, \mathbf{p}) := g \circ f_{L,\epsilon} \circ f_{L-1,\epsilon} \circ \cdots \circ f_{1,\epsilon}(\mathbf{q}, \mathbf{p}),$$

where $g(\mathbf{q}, \mathbf{p}) = (\mathbf{q}, -\mathbf{p})$ and each momentum-Verlet leapfrog step $f_{i,\epsilon}(\mathbf{q}^{[i]}, \mathbf{p}^{[i]}) = (\mathbf{q}^{[i+1]}, \mathbf{p}^{[i+1]})$ is defined by the following equations,

$$\mathbf{p}^{[i+\frac{1}{2}]} = \mathbf{p}^{[i]} - \frac{\epsilon}{2}\nabla U(\mathbf{q}^{[i]}), \quad \mathbf{q}^{[i+1]} = \mathbf{q}^{[i]} + \epsilon\mathbf{p}^{[i+\frac{1}{2}]}, \quad \mathbf{p}^{[i+1]} = \mathbf{p}^{[i+\frac{1}{2}]} - \frac{\epsilon}{2}\nabla U(\mathbf{q}^{[i+1]}), \tag{3}$$

where the *potential energy function* $U(\mathbf{q}) := -\log(\pi_{\mathbf{Q}}(\mathbf{q}))$, and $\nabla U(\mathbf{q})$ is its gradient vector. It can be easily verified that this mapping, $\mathcal{F}(\mathbf{q}, \mathbf{p})$, is reversible. Also, since $\mathcal{F}(\mathbf{q}, \mathbf{p})$ is only composed of shear mappings and a negation, it preserves the volume i.e. the absolute Jacobian determinant of the total transformation is equal to 1. As such, MHG acceptance probability (2) simplifies to:

$$\min\left\{1, \frac{\pi_{\mathbf{Q}}(\mathbf{q}^{(*)})\pi_{\mathbf{P}}(\mathbf{p}^{(*)})}{\pi_{\mathbf{Q}}(\mathbf{q}^{(c)})\pi_{\mathbf{P}}(\mathbf{p}^{(c)})} \cdot\right\} = \min\left\{1, \frac{\exp\{-(U(\mathbf{q}^{(*)}) + \frac{1}{2}\|\mathbf{p}^{(*)}\|^2)\}}{\exp\{-(U(\mathbf{q}^{(c)}) + \frac{1}{2}\|\mathbf{p}^{(c)}\|^2)\}}\right\}.$$

Evolution of a *phase state*, $(\mathbf{q}, \mathbf{p})$, via the equations of motion conserves the *Hamiltonian*, $U(\mathbf{q}) + \frac{1}{2}\|\mathbf{p}\|^2$. As such, if the leapfrog approximation is reasonably precise, HMC's proposal acceptance probability is close to 1. Seeing HMC as an instance of RJMCMC provides us with the insight required to design our own Fixed-distance HMC in the next sections.

## 3 Reversible leapfrogs that traverse a fixed distance

Designing a reversible fixed-distance leapfrog mechanism is challenging. This problem is illustrated in Figure 1 where each phase state, $(\mathbf{q}, \mathbf{p})$, is depicted by a node, $\mathbf{q}$, to which an arrow, $\mathbf{p}$, is attached. An initial phase state $(\mathbf{q}, \mathbf{p}) := (\mathbf{q}_0, \mathbf{p}_0)$ that goes through $k$ leapfrog steps (each as in Figure 1-a) evolves into $(\mathbf{q}^{[k]}, \mathbf{p}^{[k]})$ and traverses the total distance, $d^{[k]}(\mathbf{q}, \mathbf{p})$, that is:

$$d^{[k]}(\mathbf{q}, \mathbf{p}) := \sum_{i=1}^{k}\|\mathbf{q}^{[i]} - \mathbf{q}^{[i-1]}\| = \sum_{i=1}^{k}\epsilon\|\mathbf{p}^{[i-1]}\|.$$

We want to restrict the total traversed distance to a fixed value, $\mathfrak{D}$. In Figure 1-b, this distance budget is exhausted between the 4th and 5th leapfrog steps. That is, $d^{[4]}(\mathbf{q}, \mathbf{p}) < \mathfrak{D} < d^{[5]}(\mathbf{q}, \mathbf{p})$. Suppose we return $(\mathbf{q}^{[4]}, -\mathbf{p}^{[4]})$ as the final state, $\mathcal{F}(\mathbf{q}, \mathbf{p}) = (\mathbf{q}' := \mathbf{q}^{[4]}, \mathbf{p}' := -\mathbf{p}^{[4]})$. This process may seem reversible, as by 4 leapfrogs, $(\mathbf{q}^{[4]}, -\mathbf{p}^{[4]})$ is mapped back to $(\mathbf{q}, \mathbf{p})$. Nonetheless, in the backward direction, the termination condition may not happen at $(\mathbf{q}, \mathbf{p})$. For instance, in Figure 1-c, $d^{[6]}(\mathbf{q}', \mathbf{p}') < \mathfrak{D} < d^{[7]}(\mathbf{q}', \mathbf{p}')$, which suggests that $\mathcal{F}(\mathbf{q}', \mathbf{p}') = (\mathbf{q}'^{[6]}, \mathbf{p}'^{[6]})$. This violates the required RJMCMC reversibility condition. An alternative potential solution is to add an extra leapfrog step with duration, $\tau'$:

$$\tau' := \frac{\mathfrak{D} - d^{[4]}(\mathbf{q}^4, \mathbf{p}^4)}{\|\mathbf{p}^{[4]}\|} < \epsilon$$

to the forward pass which maps the state right to a point where the traversed distance, $d^{[k]}(\mathbf{q}, \mathbf{p})$, is exactly equal to $\mathfrak{D}$. That is, $\mathcal{F}(\mathbf{q}, \mathbf{p}) = (\mathbf{q}^{[x]}, -\mathbf{p}^{[4]})$. However, this process is not reversible either

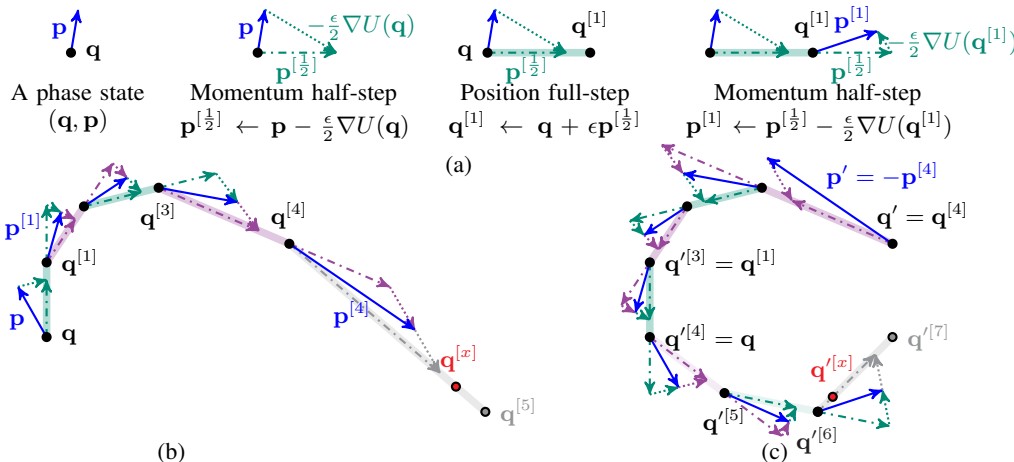

Figure 1: Fixing the traversed distance of the ordinary leapfrog mechanism: (a) A single leapfrog step mapping $(\mathbf{q}, \mathbf{p})$ to $(\mathbf{q}^{[1]}, \mathbf{p}^{[1]})$. (b) A leapfrog mechanism starts at $(\mathbf{q}, \mathbf{p})$ and exhausts the traversed distance budget between the 4th and 5th steps at $\mathbf{q}^{[x]}$. (c) The leapfrog mechanism that starts from $(\mathbf{q}' := \mathbf{q}^{[4]}, \mathbf{p}' := -\mathbf{p}^{[4]})$ exhausts the distance budget at $\mathbf{q}'^{[x]}$ and is not reversible.

because the last forward step has a duration $\tau' < \epsilon$, and does not match the first leapfrog step (of the duration $\epsilon$) in the backward direction. Our key insight is that by adding another leapfrog step of some duration, $\tau < \epsilon$, to the beginning of the process, we can make the transition symmetric and reversible. This algorithm is illustrated in Figure 2 and is described in Section 3.1.

## 3.1 Fixed-distance Leapfrog Mechanism (FD-Leapfrogs)

Given the leapfrog step size, $\epsilon$, and the total evolution distance, $\mathfrak{D}$, as hyper-parameters, we define an $(n+1)$-dimensional RJMCMC auxiliary vector $(\mathbf{p}, \tau)$ where $\tau \sim \mathrm{Unif}(0, \epsilon)$. We also define a function $\mathcal{F} : \mathbb{R}^{2n+1} \to \mathbb{R}^{2n+1}$ that maps $(\mathbf{q}, \mathbf{p}, \tau)$ to $(\mathbf{q}', \mathbf{p}', \tau')$ as follows:

**1. Initial position $\tau$-step:** Evolve the position, $\mathbf{q}$, for time $\tau$ with fixed momentum, $\mathbf{p}$.

$$\mathbf{q}^{[1]} := \mathbf{q} + \tau\mathbf{p} \tag{4}$$

**2. Initial momentum full-step:**

$$\mathbf{p}^{[1]} := \mathbf{p} - \epsilon\nabla U(\mathbf{q}^{[1]}) \tag{5}$$

**3. Interchanged position and momentum full-steps:** For $i = 2, \ldots, k$, update the position and momentum vectors interchangeably as follows:

$$\mathbf{q}^{[i]} := \mathbf{q}^{[i-1]} + \epsilon\mathbf{p}^{[i-1]} \quad (6) \qquad\qquad \mathbf{p}^{[i]} := \mathbf{p}^{[i-1]} - \epsilon\nabla U(\mathbf{q}^{[i]}) \quad (7)$$

where $k$ is the maximum number of leapfrogs s.t. the total traversed distance does not exceed $\mathfrak{D}$:

$$d^{[k]}(\mathbf{q}, \mathbf{p}, \tau) \leq \mathfrak{D} < d^{[k+1]}(\mathbf{q}, \mathbf{p}, \tau), \text{ where } \quad d^{[j]}(\mathbf{q}, \mathbf{p}, \tau) := \tau\|\mathbf{p}\| + \sum_{i=1}^{j-1} \epsilon\|\mathbf{p}^{[i]}\|, \quad \forall j \in \mathbb{N}. \tag{8}$$

**4. Final position $\tau'$-step:** With momentum $\mathbf{p}^{[k]}$, the remaining distance, $\mathfrak{d} := \mathfrak{D} - d^{[k]}(\mathbf{q}, \mathbf{p}, \tau)$, is traversed in time $\tau' := \frac{\mathfrak{d}}{\|\mathbf{p}^{[k]}\|}$, to reach the final position state, $\mathbf{q}'$:

$$\mathbf{q}' := \mathbf{q}^{[k]} + \tau'\mathbf{p}^{[k]}. \tag{9}$$

**5. Negating the final momentum:** $(\mathbf{q}', \mathbf{p}' := -\mathbf{p}^{[k]}, \tau')$ is returned as $\mathcal{F}(\mathbf{q}, \mathbf{p}, \tau)$.

**The relation between FD-leapfrogs and the ordinary leapfrog mechanism.** Traditionally, in leapfrog integration, the position and momentum vectors are updated at steps $i$ and $i + \frac{1}{2}$, respectively.

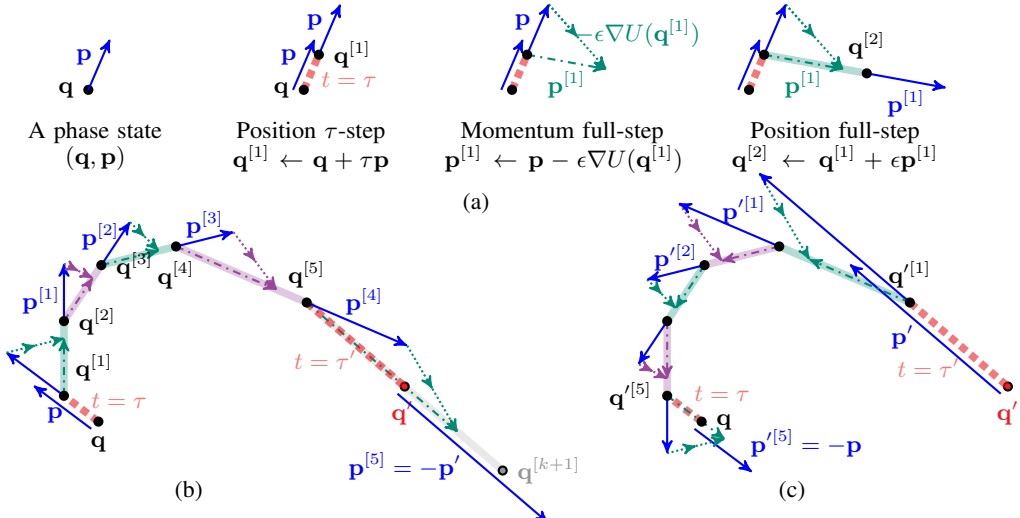

Figure 2: Reversibility of FD-leapfrogs: (a) The first 3 steps of FD-leapfrog mechanism. (b) FD-leapfrogs map $(\mathbf{q}, \mathbf{p}, \tau)$ to $(\mathbf{q}', \mathbf{p}', \tau')$. (c) FD-leapfrogs map $(\mathbf{q}', \mathbf{p}', \tau')$ back to $(\mathbf{q}, \mathbf{p}, \tau)$.

However, by shifting the step indicator by half, we end up in an equally valid leapfrog integrator that updates momentums at $i$ and positions at $i + \frac{1}{2}$, which is called *position-Verlet* integration (Batcho and Schlick, 2001) (Compare with *momentum-Verlet* (3)):

$$\mathbf{q}^{[i+\frac{1}{2}]} = \mathbf{q}^{[i]} + \frac{\epsilon}{2}\mathbf{p}^{[i]}, \qquad \mathbf{p}^{[i+1]} = \mathbf{p}^{[i]} - \epsilon\nabla U(\mathbf{q}^{[i+\frac{1}{2}]}), \qquad \mathbf{q}^{[i+1]} = \mathbf{q}^{[i+\frac{1}{2}]} + \frac{\epsilon}{2}\mathbf{p}^{[i+1]}.$$

Nonetheless, the first position update of the $i$-th leapfrog is the same as the second position update of the $(i-1)$-th leapfrog. Therefore, the ordinary position-Verlet leapfrog integration can also be formulated as: (1) an initial position evolution for duration $\frac{\epsilon}{2}$ (half-step), (2) an initial momentum full-step, (3) $(L-1)$ interchanged position and momentum full-steps, (4) a final position half-step and (5) negation of the final momentum.

Our proposed FD-Leapfrog algorithm is very similar to position-Verlet, with major differences being:

1. The durations of the initial and final position evolutions, $\tau$ and $\tau'$, are values in the interval $(0, \epsilon)$ rather than being fixed to $\frac{\epsilon}{2}$, albeit, $\mathbb{E}(\tau) = \frac{\epsilon}{2}$ since $\tau \sim \text{Unif}(0, \epsilon)$.
2. The number of interchanged position and momentum full-steps is decided based on the traversed distance rather than being fixed to $(L-1)$.

**Reversibility.** The reversibility of FD-Leapfrogs is established by Lemma 3.1 and is illustrated in the example of Figure 2. In the forward evolution (Figure 2-a), an augmented state $(\mathbf{q}, \mathbf{p}, \tau)$ is mapped to $(\mathbf{q}', \mathbf{p}', \tau')$ and the total traversed distance is:

$$\mathfrak{D} = \tau\|\mathbf{p}\| + \sum_{i=1}^{4} \epsilon\|\mathbf{p}^{[i]}\| + \tau'\|\mathbf{p}^{[5]}\|$$

(here, $k = 5$). In Figure 2-b, the evolution starts from $(\mathbf{q}', \mathbf{p}', \tau')$. It can be verified that

$$\mathbf{p}' = -\mathbf{p}^{[5]}, \ \mathbf{p}'^{[1]} = -\mathbf{p}^{[4]}, \ \dots, \ \mathbf{p}'^{[5]} = -\mathbf{p}.$$

At this stage, (i.e. $k' = 5$), the (backward) traversed distance is: $\tau'\|\mathbf{p}'\| + \sum_{i=1}^{4} \epsilon\|\mathbf{p}'^{[i]}\|$ which is equal to $\tau'\|\mathbf{p}^{[5]}\| + \sum_{i=1}^{4} \epsilon\|\mathbf{p}^{[i]}\|$. The remaining distance is: $\tau\|\mathbf{p}\|$, and by the current momentum, $\mathbf{p}'^{[5]} = -\mathbf{p}$, it is traversed in time $\tau$. Since $\tau < \epsilon$, at this stage the interchanged full-steps terminate, steps 4 and 5 of the algorithm are run and $\mathcal{F}(\mathbf{q}', \mathbf{p}', \tau') = (\mathbf{q}, \mathbf{p}, \tau)$ is returned.

**Lemma 3.1.** *Fixed-distance Leapfrog Mechanism (FD-Leapfrogs) is reversible.*[1]

Being provided with a reversible mapping $\mathcal{F}(\mathbf{q}, \mathbf{p}, \tau) = (\mathbf{q}', \mathbf{p}', \tau')$, an HMC sampler with fixed-distance evolution mechanism is proposed in Section 4.

---

[1]The formal proof is provided in the supplementary material.

## 4 Fixed-Distance Hamiltonian Monte Carlo (FDHMC)

To design a RJMCMC based on the mapping, $\mathcal{F}(\mathbf{q}, \mathbf{p}, \tau) = (\mathbf{q}', \mathbf{p}', \tau')$, we need to compute the absolute Jacobian determinant, $|\partial(\mathbf{q}', \mathbf{p}', \tau')/\partial(\mathbf{q}, \mathbf{p}, \tau)|$. The Jacobian of a complicated transformation is typically calculated via decomposing the mapping into simpler steps, as the Jacobian determinant of a composition of functions is equal to the product of the Jacobian determinants of the parts. We observe that $\mathcal{F}$ can be expressed as a composition of functions with form: $\Upsilon_i(\mathbf{q}, \mathbf{p}, \zeta) = (\mathbf{q}^*, \mathbf{p}^*, \zeta^*)$, where $\zeta$ is an auxiliary variable with alternating semantics. That is, $\zeta$ is initially set to time $\tau$, then $\zeta$ represents the remaining distance, $\mathfrak{d}$, and in the final steps, $\zeta$ represents time $\tau'$:

**Lemma 4.1.** *The Absolute Jacobian determinant of the Fixed-distance Leapfrog Mechanism (FD-leapfrogs), $\mathcal{F}(\mathbf{q}, \mathbf{p}, \tau) = (\mathbf{q}', \mathbf{p}', \tau')$, is $\frac{\|\mathbf{p}\|}{\|\mathbf{p}'\|}$.*

*Proof.* FD-Leapfrogs can be reformulated as:

$$\mathcal{F}(\mathbf{q}, \mathbf{p}, \tau) = \Upsilon_7 \circ \Upsilon_1 \circ \Upsilon_6 \circ (\Upsilon_3 \circ \Upsilon_5 \circ \Upsilon_4) \circ \ldots \circ (\Upsilon_3 \circ \Upsilon_5 \circ \Upsilon_4) \circ \Upsilon_3 \circ \Upsilon_2 \circ \Upsilon_1(\mathbf{q}, \mathbf{p}, \tau),$$

where the mappings $\Upsilon_1$ to $\Upsilon_7$ are defined as follows:

$$\Upsilon_1(\mathbf{q}, \mathbf{p}, \zeta) := (\mathbf{q} + \zeta \mathbf{p}, \mathbf{p}, \zeta), \qquad \Upsilon_2(\mathbf{q}, \mathbf{p}, \zeta) := (\mathbf{q}, \mathbf{p}, \mathfrak{D} - \zeta\|\mathbf{p}\|),$$
$$\Upsilon_3(\mathbf{q}, \mathbf{p}, \zeta) := (\mathbf{q}, \mathbf{p} - \epsilon \nabla U(\mathbf{q}), \zeta), \qquad \Upsilon_4(\mathbf{q}, \mathbf{p}, \zeta) := (\mathbf{q} + \epsilon \mathbf{p}, \mathbf{p}, \zeta)$$
$$\Upsilon_5(\mathbf{q}, \mathbf{p}, \zeta) := (\mathbf{q}, \mathbf{p}, \zeta - \epsilon\|\mathbf{p}\|), \qquad \Upsilon_6(\mathbf{q}, \mathbf{p}, \zeta) := (\mathbf{q}, \mathbf{p}, \frac{\zeta}{\|\mathbf{p}\|})$$
$$\Upsilon_7(\mathbf{q}, \mathbf{p}, \zeta) := (\mathbf{q}, -\mathbf{p}, \zeta).$$

$\Upsilon_1$ is the initial position $\tau$-step (Alg. step 1). $\Upsilon_2$ returns the remaining distance as its last output. $\Upsilon_3$ is a momentum full-step (Alg. step 2). $\Upsilon_4$ is a position full-step. $\Upsilon_5$ updates the remaining distance. As such, the composition, $(\Upsilon_3 \circ \Upsilon_5 \circ \Upsilon_4)$, stands for an interchanged position and momentum full-step, along with updating the remaining distance, $\mathfrak{d}$, (Alg. step 3) and is applied $(k-1)$ times to generate the output $(\mathbf{q}^{[k]}, \mathbf{p}^{[k]}, \mathfrak{d})$. $\Upsilon_6$ computes the remaining time, $\tau' = \mathfrak{d}/\|\mathbf{p}^{[k]}\|$. The final position $\tau'$-step (Alg. step 4) is carried out by applying $\Upsilon_1$ once more. Finally, negating the final momentum (Alg. step 5) happens in $\Upsilon_7$. Except $\Upsilon_2$ and $\Upsilon_6$, all the above transformations preserve volume (because either they are shear mappings or a negation). As such:

$$\left| \frac{\partial \mathcal{F}(\mathbf{q}, \mathbf{p}, \tau)}{\partial(\mathbf{q}, \mathbf{p}, \tau)} \right| = \left| \frac{\partial \Upsilon_6(\mathbf{q}, \mathbf{p}, \zeta)}{\partial(\mathbf{q}, \mathbf{p}, \zeta)} \right|_{(\mathbf{q}, \mathbf{p}, \zeta) \leftarrow (\mathbf{q}^{[k]}, \mathbf{p}^{[k]}, \mathfrak{d})} \cdot \left| \frac{\partial \Upsilon_2(\mathbf{q}, \mathbf{p}, \zeta)}{\partial(\mathbf{q}, \mathbf{p}, \zeta)} \right|_{(\mathbf{q}, \mathbf{p}, \zeta) \leftarrow (\mathbf{q}^{[1]}, \mathbf{p}, \tau)}$$
$$= \frac{\|\mathbf{p}\|}{\|\mathbf{p}^{[k]}\|} = \frac{\|\mathbf{p}\|}{\|\mathbf{p}'\|}. \tag{10}$$

$\square$

### 4.1 Marginal momentum distribution

If we assume that FDHM has the same momentum distribution that the existing HMC has, i.e. $\pi_{\mathbf{P}}(\mathbf{p}) := \mathcal{N}(\mathbf{0}, \mathbf{I})$, then by substituting (10) in (2), FDHMC's acceptance probability becomes:

$$\alpha\left((\mathbf{q}, \mathbf{p}, \tau) \to (\mathbf{q}', \mathbf{p}', \tau')\right) = \min\left\{ 1, \frac{\pi_{\mathbf{Q}}(\mathbf{q}') \cdot \exp(-\frac{1}{2}\|\mathbf{p}'\|^2) \cdot \mathrm{Unif}(\tau'; 0, \epsilon)}{\pi_{\mathbf{Q}}(\mathbf{q}) \cdot \exp(-\frac{1}{2}\|\mathbf{p}\|^2) \cdot \mathrm{Unif}(\tau; 0, \epsilon)} \cdot \frac{\|\mathbf{p}\|}{\|\mathbf{p}'\|} \right\}.$$

If the Hamiltonian dynamics is (roughly) exact, the above quantity is (almost) equal to $\min\left\{ 1, \frac{\|\mathbf{p}\|}{\|\mathbf{p}'\|} \right\}$. Depending on the magnitude of the current and proposed momentums, this acceptance probability can be quite low which is undesirable. Nonetheless, the interesting point that we would like to highlight is that $\alpha(\cdot)$ tends to reject proposals whose target probability is more than the target probability of the current state but always accepts the less probable proposals.[2] This rather counter-intuitive behaviour can be justified as follows: The role of the acceptance probability in the theory of MCMC is to counter-balance any bias that may exist in the proposal generation mechanism and to guarantee that the draws are proportional to the target distribution. For example, the proposal generation of a

---

[2] If $\pi_{\mathbf{Q}}(\mathbf{q}') > \pi_{\mathbf{Q}}(\mathbf{q})$, then the phase state has moved from a hill on the potential energy function into a valley and therefore, has gained more momentum. Therefore, $\|\mathbf{p}'\| > \|\mathbf{p}\|$ which leads to an acceptance probability, $\alpha(\cdot) < 1$. By a similar argument, if $\pi_{\mathbf{Q}}(\mathbf{q}') < \pi_{\mathbf{Q}}(\mathbf{q})$, then $\|\mathbf{p}'\| < \|\mathbf{p}\|$ and $\alpha(\cdot) = 1$.

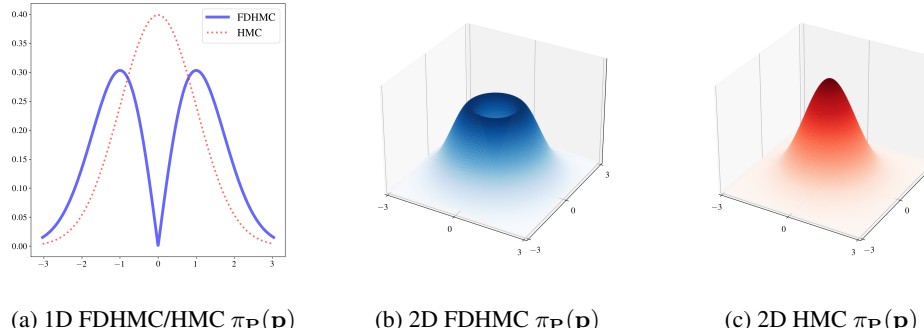

| (a) 1D FDHMC/HMC $\pi_{\mathbf{P}}(\mathbf{p})$ | (b) 2D FDHMC $\pi_{\mathbf{P}}(\mathbf{p})$ | (c) 2D HMC $\pi_{\mathbf{P}}(\mathbf{p})$ |
|---|---|---|

Figure 3: The momentum density function, $\pi_{\mathbf{P}}(\mathbf{p})$, of FDHMC versus the Gaussian momentum density function of the existing HMC associated with (a) a 1D target space, (b) & (c) 2D target spaces.

*Random Walk Metropolis-Hastings* (RWMH) sampler (Metropolis et al., 1953) is totally independent of the target density. As such, its corresponding acceptance probability, $\min\{1, \pi_{\mathbf{Q}}(\mathbf{q}')/\pi_{\mathbf{Q}}(\mathbf{q})\}$, favours the more probable proposals. The reason that the acceptance probability of HMC (with exact dynamics) is 1 is that its proposal generation mechanism is already unbiased. To be more precise, the bias towards low probable states (caused by Limitation 1) is canceled out by the bias towards high probable states (caused by Limitation 2). By fixing the distance rather than time, we have resolved the first bias. As such, the effect of the second bias prevails and is counter-balanced by $\alpha(\cdot)$. By choosing the following momentum density:

$$\pi_{\mathbf{P}}(\mathbf{p}) \propto \|\mathbf{p}\| \exp(-\|\mathbf{p}\|^2/2), \qquad \text{(FDHMC Momentum density)} \qquad (11)$$

the desirable acceptance probability of the existing HMC is re-established:

$$\alpha_{\text{FDHMC}}\left((\mathbf{q}, \mathbf{p}, \tau) \to (\mathbf{q}', \mathbf{p}', \tau')\right) = \min\left\{1, \frac{\pi_{\mathbf{Q}}(\mathbf{q}') \cdot \pi_{\mathbf{P}}(\mathbf{p}') \cdot \text{Unif}(\tau'; 0, \epsilon)}{\pi_{\mathbf{Q}}(\mathbf{q}) \cdot \pi_{\mathbf{P}}(\mathbf{p}) \cdot \text{Unif}(\tau; 0, \epsilon)} \cdot \left|\frac{\partial(\mathbf{q}', \mathbf{p}', \tau')}{\partial(\mathbf{q}, \mathbf{p}, \tau)}\right|\right\}$$

$$\stackrel{(10), (11)}{=} \min\left\{1, \frac{\pi_{\mathbf{Q}}(\mathbf{q}') \cdot \|\mathbf{p}'\| \exp(-\|\mathbf{p}'\|^2/2)}{\pi_{\mathbf{Q}}(\mathbf{q}) \cdot \|\mathbf{p}\| \exp(-\|\mathbf{p}\|^2/2)} \cdot \frac{\|\mathbf{p}\|}{\|\mathbf{p}'\|}\right\} = \min\left\{1, \frac{\pi_{\mathbf{Q}}(\mathbf{q}') \exp(-\|\mathbf{p}'\|^2/2)}{\pi_{\mathbf{Q}}(\mathbf{q}) \exp(-\|\mathbf{p}\|^2/2)}\right\}. \qquad (12)$$

Figure 3 shows that the momentum density of FDHMC is strongly inclined towards generating momentums that have higher magnitudes. In fact for FDHMC, $\pi_{\mathbf{P}}(\mathbf{0}) = 0$, while for HMC, $\pi_{\mathbf{P}}(\mathbf{0})$ is the peak of the density function. This means that FDHMC does not suffer from the second bias of HMC (i.e. Limitation 2) either, because with momentums that (in average) have higher magnitudes, the state will not be easily trapped in the valleys of the potential energy function.

## 4.2 Sampling the initial momentum

We need a fast way to directly draw a momentum (per FDHMC sample) from the distribution, $\pi_{\mathbf{P}}(\mathbf{p})$, that is defined by (11). We note that even though the support of this function is $\mathbb{R}^n$, it is totally symmetric with respect to the angular coordinates and is only a function of the magnitude, $\|\mathbf{p}\|$ (see Figure 3). Lemma 4.2 implies that the momentum magnitude, $m := \|\mathbf{p}\|$, is distributed as:

$$\pi_M(m) \propto m \cdot \exp\left(-m^2/2\right) \cdot m^{n-1} = \chi(m; n+1),$$

where $m := \|\mathbf{p}\|$ and $\chi(m; n+1) \propto m^n \exp(-m^2/2)$, is a Chi distribution defined on the support: $m \in [0, \infty)$ with the *degree of freedom*, $(n+1)$.

As such, the process of drawing a sample $\mathbf{p}^{(c)}$ from $\pi_{\mathbf{P}}(\mathbf{p}) \propto \|\mathbf{p}\| \cdot \exp\left(-\frac{1}{2}\|\mathbf{p}\|^2\right)$ is as follows:

1. Draw a direction vector, $\mathbf{v} \in \mathbb{R}^n$, from a uniform $n$-sphere, e.g. by normalising a draw from a normal distribution: $\mathbf{v} := \mathbf{s}/\|\mathbf{s}\|$ where $\mathbf{s} \sim \mathcal{N}(\mathbf{0}_n, \mathbf{I}_{n \times n})$.
2. Draw a magnitude, $m$, from a Chi distribution $\chi(m; n+1)$.
3. $\mathbf{p}^{(c)}$ is generated via scaling the drawn direction by the drawn magnitude: $\mathbf{p}^{(c)} = m \cdot \mathbf{v}$.

**Algorithm 1:** FDHMC $\left(\mathbf{q}^{(c)}, U\right)$

---

**Input:** $\mathbf{q}^{(c)}$, current state $\in \mathcal{X} \subset \mathbb{R}^n$;
$\qquad U : \mathcal{X} \to \mathbb{R}$, Potential energy function (neg log of the target density, $\pi_{\mathbf{Q}}$);
**Hyperparameters:** $\mathfrak{D} \in \mathbb{R}$, total evolution distance; $\epsilon \in \mathbb{R}$, leapfrog step-size.

---

▷ *The following line is equivalent to:* $\mathbf{p}^{(c)} \sim \pi_{\mathbf{P}}(\mathbf{p}) \propto \|\mathbf{p}\| \exp(-\frac{\|\mathbf{p}\|^2}{2})$
$\mathbf{s} \sim \mathcal{N}(\mathbf{0}_n, \mathbf{I}_{n \times n}); \; m \sim \chi(m; n+1); \; \mathbf{p}^{(c)} \leftarrow m \cdot \frac{\mathbf{s}}{\|\mathbf{s}\|}$ ▷ *where* $\chi(m; n+1) \propto m^n \exp(-m^2/2)$
$\tau^{(c)} \sim \text{Unif}(0, \epsilon)$          ▷ *duration of the initial fixed-momentum evolution*
$\mathbf{q} \leftarrow \mathbf{q}^{(c)} + \tau^{(c)} \cdot \mathbf{p}^{(c)}$       ▷ *initial evolution of the position vector for time* $\tau^{(c)}$
$\mathfrak{d} = \mathfrak{D} - \tau^{(c)} \cdot \|\mathbf{p}^{(c)}\|$         ▷ *This is the remaining distance*
$\mathbf{p} \leftarrow \mathbf{p}^{(c)} - \epsilon \nabla U(\mathbf{q})$       ▷ *initial evolution of the momentum vector for time* $\epsilon$

▷ *while a position full-step evolution is possible without exceeding the remaining distance,* $\mathfrak{d}$:
**while** $\epsilon \cdot \|\mathbf{p}\| < \mathfrak{d}$ **do**
 |   $\mathbf{q} \leftarrow \mathbf{q} + \epsilon \cdot \mathbf{p}$           ▷ *position full-step*
 |   $\mathfrak{d} \leftarrow \mathfrak{d} - \epsilon \cdot \|\mathbf{p}\|$        ▷ *update the remaining distance*
 |   $\mathbf{p} \leftarrow \mathbf{p} - \epsilon \nabla U(\mathbf{q})$        ▷ *momentum full step*

$\tau^{(*)} \leftarrow \frac{\mathfrak{d}}{\|\mathbf{p}\|}; \quad \mathbf{q}^{(*)} \leftarrow \mathbf{q} + \tau^{(*)} \cdot \mathbf{p}; \quad \mathbf{p}^{(*)} \leftarrow -\mathbf{p};$   ▷ *constructing the final proposal*

**if** $u \sim \text{Unif}(0, 1) \; < \; \frac{\pi_{\mathbf{Q}}(\mathbf{q}^{(*)}) \exp\left(-\|\mathbf{p}^{(*)}\|^2/2\right)}{\pi_{\mathbf{Q}}(\mathbf{q}^{(c)}) \exp\left(-\|\mathbf{p}^{(c)}\|^2/2\right)}$ **then return** $\mathbf{q}^{(*)}$ **else return** $\mathbf{q}^{(c)}$

---

**Lemma 4.2.** *If a density function,* $\pi_{\mathbf{X}}(\mathbf{x}) := \pi_{\mathbf{X}}(x_1, \ldots, x_n)$ *(on* $\mathbb{R}^n$*), is only a function of magnitude,* $\|\mathbf{x}\|$*, then the random variable* $m = \|\mathbf{x}\|$*, is distributed with density* $\pi_M(m) \propto m^{n-1} \cdot g(m)$*, where* $g(m) := \pi_{\mathbf{X}}([m, 0, \ldots, 0]^\top) = \pi_{\mathbf{X}}([0, m, 0 \ldots, 0]^\top) = \ldots = \pi_{\mathbf{X}}([0, \ldots, 0, m]^\top)$.

*Proof.* The CDF, $\Pi_M(a)$, meaning the probability that $m < a$ is:

$$\Pi_M(a) := \int_{m=0}^{a} \pi_M(m) dm. \tag{13}$$

This is equivalent to integrating over the density function, $\pi_{\mathbf{X}}(\mathbf{x})$, within the hypersphere, $\|\mathbf{x}\| \leq m$:

$$\Pi_M(a) = \int_{x_1=-\infty}^{\infty} \ldots \int_{x_n=-\infty}^{\infty} \mathbb{I}\left[\|\mathbf{x}\| \leq a\right] \pi_{\mathbf{X}}(\mathbf{x}) d\mathbf{x}, \quad \text{integration on an } n\text{-sphere of radius } a$$

$$= \int_{r=0}^{a} \int_{\phi_1=0}^{\pi} \ldots \int_{\phi_{n-2}=0}^{\pi} \int_{\phi_{n-1}=0}^{2\pi} g(r) \cdot r^{n-1} \sin^{n-2}(\phi_1) \sin^{n-3}(\phi_2) \ldots \sin(\phi_{n-1})$$

$$dr d\phi_1 \ldots d\phi_{n-1}, \text{ with Jacobian terms due to the change of coordinates to spherical}$$

$$\propto \int_{r=0}^{a} g(r) \cdot r^{n-1} dr. \tag{14}$$

Comparing (13) and (14) shows that $\pi_M(m) \propto g(m) \cdot m^{n-1}$.       □

The complete FDHMC sampler is illustrated in Algorithm 1. It can be seen that in terms of implementation complexity, this algorithm is quite comparable with the baseline HMC. Nonetheless, as our experimental results will show, in terms of *Effective Sample Size* per gradient (ESS/grad), FDHMC can largely outperforms the baseline.

## 5 Experimental evaluation

We compare the performance of FDHMC against 3 baselines: (a) static HMC (Neal, 2011), and two versions of No-U-turn sampler (Hoffman and Gelman, 2014) (NUTS) including (b) *Dynamic Slice HMC* (DSHMC), which is the original implementation (Hoffman and Gelman, 2014), and (c) *Dynamic Multinomial HMC* (DMHMC) (Betancourt, 2017), which is the current version of NUTS used in STAN (Carpenter et al., 2017).

For tuning HMC, the initial total simulation duration, $\lambda = \epsilon L$, is set to 2.0 and the step-size is tuned by dual averaging (Nesterov, 2009; Hoffman and Gelman, 2014). If dual averaging does not converge

we halve the value of $\lambda$ and repeat this process until the convergence conditions are satisfied. For tuning DSHMC and DMHMC, we rely on *Mici* probabilistic programming language (Graham, 2019).

We tune the the parameters of FDHMC by an algorithms that is presented in the supplementary material and works well on our experimental models. This includes tuning the step-size, $\epsilon$, via dual averaging and the following heuristic for tuning the fixed distance, $\mathfrak{D}$: We observe that if $\mathfrak{D}$ is too large, its correlation with the expected distance of the successive samples, $\mathbb{E}[\|\mathbf{q}^{(i)} - \mathbf{q}^{(i-1)}\|]$, is low. As such, we primarily draw $N = 500$ samples, $\mathbf{q}^{(i)}$, from an FDHMC sampler that is associated with a sufficiently large fixed-distance, $\mathfrak{D}^*$. Then, we set

$$\mathfrak{D} = \frac{1}{N-1} \sum_{i=2}^{N} \|\mathbf{q}^{(i)} - \mathbf{q}^{(i-1)}\|,$$

as the final fixed-distance. To decide the initial $\mathfrak{D}^*$, we set $\epsilon^* = 1$ and then repeatedly double or halve the value of $\epsilon^*$ till the acceptance probability of the Langevin proposal with step-size $\epsilon^*$ crosses 0.5. Then we set $\mathfrak{D}^* := 10\epsilon^*$. To stabilise the performance of this Langevin dynamics, the magnitude of its associated momentum vector, $\mathbf{p}$, is fixed to the mean of $\chi(x, n+1)$, i.e.

$$\|\mathbf{p}\| := \sqrt{2}\Gamma(n/2 + 1)/\Gamma((n+1)/2).$$

Our experimental models include:

1. $n$-dimensional multivariate normal distributions (MVN) where $n \in \{10, 30, 100, 300\}$ and the covariance matrix is drawn from a Wishard distribution which has an identity scale matrix and its *degree of freedom* is equal to the model's dimension, $n$.
2. Neal's funnel (Neal, 2003) (FNNL):

$$\pi_{\mathbf{Q}}(q_1, \ldots, q_n) = \mathcal{N}\left(q_1; 0, \sigma^2\right) \prod_{i=2}^{n} \mathcal{N}\left(q_i; 0, \exp(kq_1)\right),$$

   where the dimension, $n \in \{5, 10, 50, 100\}$, $\sigma^2 = 1$ and $k = 3$.
3. The posterior density function, $\pi_{\mathbf{Q}}(\alpha, \boldsymbol{\beta}|X, \mathbf{y})$ (15), associated with Bayesian Logistic Regression for Binary classification where each data point contains a vector of predictors, $\mathbf{x}_i$, and the class label $\mathbf{y}_i \in \{-1, 1\}$:

$$\pi_{\mathbf{Q}}(\alpha, \boldsymbol{\beta}|X, \mathbf{y}) \propto \exp\left(-\frac{\alpha^2}{2\sigma^2} - \frac{\boldsymbol{\beta}^\top \boldsymbol{\beta}}{2\sigma^2} - \sum_{i=1}^{\#\text{data}} \log\left(1 + \exp(-y_i(\alpha + \mathbf{x}_i \cdot \boldsymbol{\beta}))\right)\right). \quad (15)$$

   We set $\sigma^2 = 1$ and apply the model to the following three data sets that are available from the UCI repository (Frank et al., 2011): (a) *Australian Credit Approval* (AusCr) data set where each data point have 14 predictors (as such, $n = 15$) and we only use the first 100 data points in the data set. (b) *SPECT Heart* (SPECT) data set with 267 data points, each with 22 predictors. (c) *German Credit* (GrCr) data set with 1000 data points, each with 24 predictors. In these 3 data sets, we normalise all predictors, to have zero mean and unit variance.

For each sampling algorithm, 50 independent MCMC chains are run and the average *Effective Sample Size per gradient* (ESS/grad) $\pm$ 95% condifence interval is reported in Table 1. The length of each chain is 1000 samples which are drawn after an initial 200 burn-in draws. In each chain iteration, all samplers start from a same initial states and the ESS is approximated by following the details described by Hoffman and Gelman (2014).

The results show that in 8 out of the 11 experiments, FDHMC outperforms the baselines and in some models the difference is substantial.

## 6  Conclusion

The main insight of the present work is that the fixed-time evolution of a phase state via the equations of motion is intrinsically biased towards generating proposals that have low target probability (Limitation 1). In HMC, this bias is compensated by drawing momentums from a normal distribution centered at $\mathbf{0}$. Many such sampled momentum vectors have small magnitudes and do not let the

| Model | dim | HMC | DSHMC | DMHMC | FDHMC |
|---|---|---|---|---|---|
| MVN | 10 | $0.0174 \pm 0.0026$ | $\mathbf{0.0318 \pm 0.0007}$ | $0.0308 \pm 0.0005$ | $0.0252 \pm 0.0036$ |
| MVN | 30 | $0.0006 \pm 0.0002$ | $0.0012 \pm 0.0001$ | $0.0012 \pm 9.80e^{-5}$ | $\mathbf{0.0031 \pm 0.0025}$ |
| MVN | 100 | $0.0002 \pm 0.0001$ | $0.0002 \pm 2.29e^{-5}$ | $0.0002 \pm 2.39e^{-5}$ | $\mathbf{0.0591 \pm 0.0123}$ |
| MVN | 300 | $0.0069 \pm 0.0031$ | $0.0017 \pm 8.39e^{-6}$ | $0.0015 \pm 6.78e^{-6}$ | $\mathbf{0.0629 \pm 0.0069}$ |
| FNNL | 5 | $\mathbf{0.0084 \pm 0.0023}$ | $0.0006 \pm 0.0001$ | $0.0011 \pm 0.0002$ | $0.0080 \pm 0.0026$ |
| FNNL | 10 | $0.0027 \pm 0.0009$ | $0.0005 \pm 0.0001$ | $0.0005 \pm 0.0001$ | $\mathbf{0.0030 \pm 0.0012}$ |
| FNNL | 50 | $0.0007 \pm 0.0005$ | $0.0001 \pm 5.72e^{-5}$ | $8.18e^{-5} \pm 2.32e^{-5}$ | $\mathbf{0.0008 \pm 0.0003}$ |
| FNNL | 100 | $0.0009 \pm 0.0005$ | $8.36e^{-5} \pm 3.61e^{-5}$ | $9.11e^{-5} \pm 3.87e^{-5}$ | $\mathbf{0.0045 \pm 0.0078}$ |
| AusCr | 15 | $0.0132 \pm 0.0010$ | $\mathbf{0.0286 \pm 0.0011}$ | $0.0276 \pm 0.0008$ | $0.0241 \pm 0.0077$ |
| SPECT | 23 | $0.0122 \pm 0.0008$ | $0.0190 \pm 0.0004$ | $0.0254 \pm 0.0006$ | $\mathbf{0.0259 \pm 0.0013}$ |
| GrCr | 25 | $0.0003 \pm 0.0001$ | $0.0225 \pm 0.0009$ | $0.0241 \pm 0.0007$ | $\mathbf{0.0252 \pm 0.0016}$ |

Table 1: Comparing FDHMC with HMC, DSHMC and DMHMC, based on the Effective Sample Size per gradient, that is approximated via 50 independent MCMC chains, each of 1000 states.

state leave the high-probability regions. This causes a (second) bias that compensates the first bias. Nevertheless, this counterbalancing is obtained in the expense of more correlation between the successive states that are drawn from the high probability regions (Limitation 2). In the light of RJMCMC theory, we modified the core of the HMC algorithm and designed a sampler where the state is evolved for a fixed traversed distance rather than a fixed evolution time. This dynamics is not biased towards the proposals that have low target probability and is coupled with a momentum distribution that generates vectors which have a higher expected magnitude. According to our experimental results, this can lead to a better exploration of the space and a higher effective sample size per gradient while maintaining the high proposal acceptance probability of the baseline HMC.

The proposed FDHMC can be implemented efficiently and with appropriate modifications, it can be combined with many variations of HMC. More specifically, even though for simplicity throughout we focused on combining FD-leapfrogs with the baseline HMC, this mechanism can be combined with any variation of HMC that relies on Stormer–Verlet leapfrog integration e.g. the case where the mass matrice is not unit. Studying the performance of such variations of FDHMC as well as alternative methods to tune its hyper-parameters can be pursued in a future line of research.

## Acknowledgments and Disclosure of Funding

This research has been supported by the *National Health and Medical Research Council* (NHMRC), Research Grant: GNT1149976.
PURL: http://purl.org/au-research/grants/nhmrc/GNT1149976.

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
