# OpenReview forum: "Fixed-Distance Hamiltonian Monte Carlo"
_NeurIPS.cc/2022/Conference — NeurIPS 2022 Accept_

### Official Review · Reviewer_d8Ux · 2022-07-09

**Rating:** 6
**Confidence:** 4
**Soundness:** 3 good
**Presentation:** 4 excellent
**Contribution:** 4 excellent

**Summary:**

The paper proposes fixed distance Hamiltonian Monte Carlo (FDHMC), a variation of HMC that simulates the equation of motion for a fixed distance rather than a fixed amount of time. The paper establishes the theoretical foundations of FDHMC, and demonstrates its advantages over HMC and NUTS with numerical experiments.

**Questions:**

See above

**Limitations:**

The main limitation I see is the potential challenge in tuning the hyperparameters involved in FDHMC. The authors have made some discussions on how to address this limitation. But the discussions are not entirely convincing since the baseline results involving HMC and NUTS do not seem convincing. It would be ideal if the authors can:

- Establish baseline results involving HMC and NUTS using more thoroughly tested PPL packages, using the provided parameter tuning scheme (e.g. for NUTS which is fully automated in most cases).
- Study how sensitive the performance of FDHMC is to the choice of hyperparameters (e.g. if with some simple heuristics FDHMC can already outperform HMC and NUTS then the FDHMC performance is probably robust).
- Demonstrate how the automated parameter tuning presented in the supplementary can further improve performance.


**Strengths And Weaknesses:**

## Strengths

- The idea is novel, and the theoretical foundations are clearly presented with excellent organization of the materials.
- The proposed FDHMC algorithm makes intuitive sense, and can be easily implemented efficiently. This means FDHMC has the potential to make a big impact due to the widespread applications of HMC.

## Weaknesses

 While the theoretical foundations are excellently presented, I have some concerns regarding the numerical evaluation. While the motivation for FDHMC makes sense intuitively, I would like to see more evidence from numerical experiments to confirm the advantages of FDHMC over HMC/NUTS:

- In the experimental results in Section 5, vanilla HMC outperforms NUTS in many cases, sometimes by a very large margin. This seems contradictory to many of the existing results. The authors seem to indicate this is due to a simple tuning heuristic for HMC (L242-243) but I find this claim to be strange and surprising. Given the experiments are done using customized implementations and NUTS is quite a complicated algorithm, the results do not seem convincing to me. I would like to ask the authors to reimplement some of the experiments using readily available and more thoroughly tested probabilistic programming language (PPL) packages (e.g. NumPyro which has a NUTS implementation where all parameter tuning are automated and readily available tutorials on models like the Neal's funnel) and re-verify that HMC still outperforms NUTS.
- The paragraph on parameter tuning is quite confusing. Why is the total simulation duration in HMC chosen to be 2 (L239-240)? What is 2 here, and why is this number independent of the model? For FDHMC, there is some discussion on how to pick the fixed distance, but what about the step size? Is it picked the same way as HMC? What would happen if we apply some of the automated parameter tuning utilities available in commonly used PPL packages?
- In the supplementary materials the authors present an algorithm to automatically tune the hyperparameters involved in FDHMC. Is there any reason why this algorithm is not used for the experiments in the main paper?

I would be happy to bump up my score if the above concerns regarding evaluation can be properly addressed, but in its current form I still have concerns over accepting the paper.

---

> ### Author Response · Authors · 2022-08-02
> **Responses to Reviewer d8Ux**
>
> We would really like to thank the reviewers for their comments. Addressing the raised issues has surely increased the quality of the paper.
>
> * * * * *
> Reviewer d8Ux
>
> COMMENT 1. On concerns about the implementation of NUTS:
>
> Response: We thank the reviewer for highlighting this issue. As the reviewer suggested, now we use Mici (https://github.com/matt-graham/mici) implementation of NUTS. Mici is a PPL that can easily be combined with Python code. It provides two implementations of NUTS: (1) Dynamic Slice HMC (which is the original NUTS algorithm) and (2) Dynamic Multinomial HMC (which is a variation that is used in STAN since 2017 and is reported to outperform the original NUTS). We configure these samplers with Mici’s automated tuning. In average, these implementations perform better than our implementation of NUTS and their performance is more consistent with static HMC.
> As such, now in the main text, we compare FDHMC versus Mici’s implementations of NUTS. This does not affect the comparative results and FDHMC still outperforms the baseline. (see the experimental results section in the updated main text).
>
>
> COMMENT 2: Why is the total simulation duration in HMC chosen to be 2? What is 2 here, and why is this number independent of the model?
> Response.
> We are not aware of any systematic way to tune the simulation length (lambda=epsilon*L) of HMC (apart from choosing L dynamically as in NUTS). As such we chose this value by try and error and partly based on the quantitative analysis provided in the original NUTS paper.
> Note that we use this value as an upper limit. Because if the dual averaging (by which epsilon is tuned) does not converge, we keep halving lambda until the algorithm converges. This happens in some of our models. Therefore, the chosen simulation duration is not entirely independent of the model.
>
>
>
> COMMENT 3: For FDHMC, there is some discussion on how to pick the fixed distance, but what about the step size? Is it picked the same way as HMC? What would happen if we apply some of the automated parameter tuning utilities available in commonly used PPL packages?
>
> Response. Yes, the leapfrog step size of all algorithms that we have reported on (i.e. HMC, NUTS and FDHMC), is tuned by dual averaging. Dual averaging is by far the most popular way to tune the step-size of all HMC-based algorithms. Note that different samplers require slightly different variations of dual averaging (e.g. the objective function that NUTS’ dual averaging maximises is different from that of the static HMC). To make a dual averaging algorithm that is suitable for FDHMC we modified the HMC’s dual averaging (compare Algorithms 2 and 3 in the presented supplementary material with Algorithms 4 and 5 in the original NUTS paper). FDHMC’s dual averaging is a component of the fully automated FDHMC tuning which is already utilised in all experiments of the main text.
> We could apply an existing automated parameter tuning to FDHMC, but we needed to have access to the source code and modify/customise it to work with the fixed-distance dynamics.
>
> COMMENT 4. In the supplementary materials the authors present an algorithm to automatically tune the hyperparameters involved in FDHMC. Is there any reason why this algorithm is not used for the experiments in the main paper?
>
> Response. This should have been a misunderstanding. What is presented in the supplementary material is the details of the hyper-parameter tuning algorithm that we have used in the experiments of the main paper. This includes the heuristic to choose the fixed-distance as well as a dual-averaging algorithm for tuning FDHMC’s step-size.
>
> COMMENT 5. Study how sensitive the performance of FDHMC is to the choice of hyperparameters
>
> Response:
> To address this issue, we have added a new Quantitative Analysis section to the supplementary material. In this section we study the sensitivity of FDHMC to the fixed-distance parameter and show that the proposed automated parameter tuning performs reasonably well and chooses parameter values that are not far from the optimal range.
>
> COMMENT 6. Demonstrate how the automated parameter tuning presented in the supplementary can further improve performance.
>
> Response:
> (1)  As mentioned, the experimental results of the main text already rely on this automated parameter tuning. Now we have modified the first paragraph of the experiments section (in the main text) to convey this more clearly and prevent confusions.
> (2) We have studied the performance of FDHMC’s automated tuning in the newly added quantitative analysis section in the (updated) supplementary material.

---

> > ### Comment · Reviewer_d8Ux · 2022-08-07
> > **Thanks for the response**
> >
> > I thank the authors for the response, and for the added experiments and the updated manuscript.
> >
> > The updated results with MICI make more sense to me (although I have to say that I find it quite concerning that the updated results differ so much from the previous results: many cases have 2x or even 5x improvements with the NUTS implementation from MICI). Please also update the rest of the manuscript to be consistent with the change (e.g. updated supplementary L111, attach updated code with MICI).
> >
> > I have updated my score accordingly.

---

> > > ### Author Response · Authors · 2022-08-08
> > > **Code Updated**
> > >
> > > We thank the reviewer for suggesting using an existing PPL.
> > > The poor performance of our implementation of NUTS relative to the static HMC had been concerning for us too. But after spending a substantial time on debugging the code and finding no issue, we were confident that the results are correct (despite being unexpected).  Mici's results have surprised us again. There is a possibility that the difference is only due to a more effective parameter tuning.  However, as the reviewer suspected, it can also be due to an undiscovered bug in our implementation or tuning of NUTS. In any case, now we only report on Mici's implementations. Also, we updated the attached code with MICI's samplers.

---

### Official Review · Reviewer_zous · 2022-07-11

**Rating:** 6
**Confidence:** 4
**Soundness:** 3 good
**Presentation:** 3 good
**Contribution:** 3 good

**Summary:**

The paper proposes a new Hamiltonian Monte Carlo algorithm for probabilistic inference in which each step of the proposal simulates the Hamiltonian dynamics for a fixed distance traversed in state space rather than a fixed number of steps. The paper motivates this alternate algorithm and provides the details of the algorithm and a proof of correctness. Empirical results comparing the new FDHMC algorithm with classical HMC and NUTS are also provided showing improved effective sample size per gradient estimation.

The motivation for the new FDHMC algorithm (paraphrasing the authors) is that the existing HMC algorithm traverses through high probability regions with very high momentum and thus doesn't spend enough time in these regions and when it does stop in the high probability regions it tends to get stuck there producing highly correlated samples.


**Questions:**

How would one incorporate learning of the mass matrix into FDHMC? If the mass matrix is not unit does this pose any complications in terms of calculations of the remaining distance?

Doesn't NUTS address Limitation 2?

I would suggest to the authors that they pick some model with a complex multimodal posterior where each mode has roughly equal probability mass. If they can show that FDHMC explores all the modes better than HMC/NUTS then one could argue that it is addressing the limitations that are claimed to be addressed.

Another minor suggestion would be to include some results on a hierarchical model. These are customary to show in papers on probabilistic inference :)

Minor Points:

- I would suggest to use a different letter other than `p` for the superscript of `q` on lines 87, for example, and in related equations. `p` is used for momentum elsewhere and this is a bit confusing.

- In equation 2 the Jacobian should refer to $\cal{F}$ and I feel that the variables should be q,r and in the partial derivative rather than q and p.


**Limitations:**

Not aware of any negative societal impact of this work.

**Strengths And Weaknesses:**

The proposed FDHMC algorithm certainly appears novel. [This reviewer has read and reviewed a large number of variants of HMC but not quite this one!]

Most current work on improving HMC algorithms seem to be focused on improving the curvature of the state space which appears to be a critical challenge in probabilistic inference using any kind of gradient-based methods. This paper doesn't address the curvature issue. The proposed FDHMC algorithm assumes a unit mass for the momentum. All of the examples in the evaluation seem to be selected such that unit mass matrix would work well for them. Hence the the paper scores low on relevance.

In terms of quality and clarity the paper is very well written and the mathematical proofs and algorithms are very easy to follow.

The two main motivations of the paper Limitation 1 and 2 are presented somewhat informally. I am able to follow the main arguments that the authors are making, but still I would have preferred to see a more mathematical justification. At the very least the empirical evaluation should somehow directly tie in the higher ESS numbers to these limitations somehow. Otherwise it could well be that the higher ESS numbers are explained by something else entirely. For example, the tuning methodology of distance traversed could have a huge influence. (As we can see in the paper that the tuning of HMC seems to produce even better results than NUTS!)

Also, while the experimental results demonstrate higher ESS numbers they don't show that the samples are exploring the high probability region of the posterior space. It might be worth showing the unnormalized posterior probabilities of the samples. In the case of logistic regression it is also customary to show the log likelihood of held out test data using the posterior samples of $\alpha$ and $\beta$.

---

> ### Author Response · Authors · 2022-08-02
> **Responses to Reviewer zous**
>
> We would really like to thank the reviewers for their comments. Addressing the raised issues has surely increased the quality of the paper.
>
>
> * * * * *
> Reviewer zous
>
> COMMENT 1. How would one incorporate learning of the mass matrix into FDHMC? If the mass matrix is not unit does this pose any complications in terms of calculations of the remaining distance?
>
> Response:  We thank the reviewer for highlighting this point. For simplicity and clarity of the discussion, in this work we focused on combining the baseline HMC with the fixed-distance mechanism. However, this approach is vastly generalisable and combining Fixed-distance Leapfrog Mechanism with any variation of HMC that relies on Stormer–Verlet leapfrog integrator is straight forward (this is a very important point, as such we have now mentioned it in the last sentence of the revised conclusion). This includes the incorporation of mass matrices as well: If the position vector, $q$, is updated as: $q = q + u$ (where $u$ is an update vector e.g. $u= \epsilon \cdot M^{-1} \cdot p$, with $M$ being a mass matrix), then the remaining distance, $D$, is simply updated as: $D = D – ||u||$.  The physical interpretation is as follows: the traversed distance is the evolution-time multiplied by the magnitude of the velocity vector. In our paper, velocity and momentum were a same vector (since momentum = mass $\times$ velocity) but if mass, in not unit, the velocity vector is the momentum vector divided by mass. That is, $M^{-1} \cdot p$ if mass, $M$, is a matrix.
>
>
> COMMENT 2. Doesn't NUTS address Limitation 2?
>
> Response: No, it does not, because similar to the static HMC (and any existing variation of HMC that we are aware of), the distribution of NUTS’ momentum is Gaussian. Also, note that Limitations 1 & 2 can only be addressed together (since the opposite biases that they induce, nullify each other).
>
> COMMENT 3. I would suggest to the authors that they pick some model with a complex multimodal posterior where each mode has roughly equal probability…
>
> Response: We thank the reviewer for this suggestion. We have added a new section to the supplementary material where FDHMC and other samplers are run on a mixture of 4 Gaussian distributions with different dimensions. The sample plots shows that FDHMC’s transitions between the distribution modes occurs much more frequently. Given that the results are quite interesting and reveal the power of Fixed-distance mechanism, we plan to add a Mixture-of-multivariate-normal-densities model to the experiment of the main text, in the camera-ready submission.
>
> COMMENT 4. Minor suggestions:
>
> Response: Thanks for highlighting the confusing notation and the typo. They are fixed now. Unfortunately, the time limitation did not allow us to implement other suggested experiments, but we will consider reporting on hierarchical models with hold-out test data in the camera-ready version.

---

### Official Review · Reviewer_U811 · 2022-07-12

**Rating:** 6
**Confidence:** 4
**Soundness:** 4 excellent
**Presentation:** 4 excellent
**Contribution:** 3 good

**Summary:**

The authors proposed a new way of implementing the Hamiltonian Monte Carlo method. This new approach restricts the distance a single HMC step will travel, hence to avoid unnecessary oscillations in low probability region, as well as to reduce the correlations between samples. The proposed algorithm has been tested against several well-known examples for HMC, and compared with existing method. IT can be seen that the proposed algorithm outperform the pain HMC, as well as NUTS(no-U-turn-sampling).

**Questions:**

1. Please provide the argument for the statement of "the expected (rather than exact) step of the initial and final positions is at $i+\epsilon/2$. (L151-L152).

2. How does the fixed-distance change for each example, since it is determined by the procedure discussed in the first paragraph of Sec. 5. Also how sensitive is the performance with respect to the distance parameter?



**Limitations:**

yes

**Strengths And Weaknesses:**

Strengths: The idea is innovative and effective in dealing with the limitations the author discussed. The algorithm is carefully designed to ensure reversibility while achieving fixed-distance traveling. The numerical experiments are representative and convincing.

Weaknesses: Quantitative analysis of the algorithm is lacking. For example, it is curious to know how sensitive the performance of the FDHMC algorithm is with respect to the parameter of the fixed-distance,

---

> ### Author Response · Authors · 2022-08-02
> **Responses to Reviewer U811**
>
> We would really like to thank the reviewers for their comments. Addressing the raised issues has surely increased the quality of the paper.
>
> * * * * *
> Reviewer U811
>
> COMMENT 1. Quantitative analysis of the algorithm is lacking. How sensitive is the performance with respect to the distance parameter?
>
> Response: Thanks a lot for mentioning this! To address this issue, now we have added a new Quantitative Analysis section to the supplementary material. In this section, we have configured FDHMC with a range of fixed-distance parameters and plotted the ESS/grad versus the fixed-distance for all the experiments of the main text.  (The python script that generates these results is added to the code in the supplementary material).
>
>
> COMMENT 2. How does the [automatically tuned] fixed-distance change for each example?
>
> Response: To address this question we have added another table in the mentioned new Quantitative Analysis section. In this table we report the average automatically tuned fixed-distance for each experiment in the main text. This value varies a lot for different models (from 0.8 to 6.5).  Comparing this table with the sensitivity plots indicates that the proposed tuning mechanism is reasonably effective, and the tuned fixed-distance is not far from the peak of ESS/grad-vs-fixed-distance curve.
> (The python script that generates the entries of this table is added to the code in the supplementary material).
>
> COMMENT 3. Argument for the statement of ‘the expected (rather than exact) step of the initial and final positions…’ at . (L151-L152).
>
> Response: We thank the reviewer for highlighting this sentence. The precise statement would be to say: In FDHMC, the expected duration of the first position update is epsilon/2, because it is drawn from Unif(0, epsilon), and the duration of the last FDHMC position update is between 0 and epsilon (as otherwise, another full-step evolution would be added).
> Computing the expected value of the last position update is non-trivial and is not required for our discussion. Now we have edited/re-written the problematic paragraph in the revised version.

---

> ### Comment · Reviewer_U811 · 2022-08-09
> **Final comment**
>
> I appreciate the authors' responses related to the questions. While I can see the materials provided have offered more evidence on the performance of FDHMC. these numerical evidences are not as convincing as a thorough and theoretical study of the algorithm. Therefore, my rating remain unchanged.

---

> > ### Author Response · Authors · 2022-08-09
> > **Asking final advice**
> >
> > Once again, thanks for your suggestions and time spent on reviewing our paper.
> > While we can still communicate, in case you have a particular numerical experiment in mind or could refer us to a related theoretical study that would potentially add to the value of the paper, we would appreciate if you could share it with us.

---

### Meta-Review · Area_Chair_WQNm · 2022-08-26

**Recommendation:** Accept
**Confidence:** Certain

**Metareview:**

This paper proposes a new variant of Hamiltonian Monte Carlo. Rather than using a fixed number of iterations (as in the original HMC) or choosing the step-size adaptively (as in NUTS) the paper simulated the dynamics until a fixed *distance* has been traversed. The paper gives some arguments why this might be a good idea and a careful proof of detailed balance for thew new proposed algorithm. Reviewers agreed the algorithm seemed correct and the numerical results were compelling, but there were some existing concerns about the implementation of baseline algorithms and some questions about the technical details of the algorithm. Given the importance of HMC and the novelty and agreed correctness of this work, I recommend acceptance and urge the authors to consider the clarifying questions asked by the reviewers as opportunities for improving the paper. Also, additional evidence for experiments (e.g. perhaps comparing to another implementation of NUTS/HMC) would also be helpful.

**Award:**

No

---

### Decision · Program_Chairs · 2022-09-14

Accept